behaviour/cognition/psychology

negative affective bias, affective bias modification, mood, mood disorders, effectiveness research, prevention

**Author for correspondence:**
Alysha Chelliah
e-mail: alysha.chelliah@gmail.com

# Efficacy of attention bias modification via smartphones in a large population sample

## Alysha Chelliah and Oliver Robinson

Neuroscience and Mental Health Group, Institute of Cognitive Neuroscience, University College London, 17-19 Queen Square, London WC1N 3AZ, UK

AC, 0000-0003-0867-1565

Negative affective biases are a key feature of anxiety and depression that uphold and promote negative mood. Bias modification aims to reduce these biases using computerized training, but shows mixed success and has not been tested at scale. The aim was to determine whether bias modification delivered via smartphones can improve mood in a large sample. In total, 153 385 self-referring participants were randomly assigned to modification or sham bias training on a dot-probe or visual-search task. The primary outcome of interest was balance of mood, assessed on the Positive and Negative Affect Schedule. In total, 22 933 participants who provided at least two mood ratings were included in analyses. There was a large amount of participant attrition. In the remaining smaller sample, results supported the prediction that visual-search modification would result in improved mood (95%CI [0.10, 0.82]; $p = 0.01$, $d = 0.05$, $N = 2588$ after two ratings; 95%CI [1.75, 6.54]; $p = 0.001$, $d = 0.32$, $N = 118$ after six ratings), which was not seen for the sham version ($N = 4818$ after two ratings; $N = 138$ after six ratings). Dot-probe modification was not associated with mood improvements ($p = 0.52$). Visual-search, but not dot-probe, bias modification slightly but significantly improved mood. Although this effect size is very small and subject to large participant drop-off, it might be worth considering an adjunct to current treatments.

## 1. Introduction

Negative affective bias is the tendency to preferentially process negative emotional information [1] including preferential attention towards negative stimuli [2]. It plays a critical causal role in onset and maintenance of negative mood, and in turn of anxiety disorders and depression [1,3–6]. These affective

disorders pose a significant global concern [7], but there is a lack of effective, affordable and readily accessible interventions [8–12].

The ability to modify negative affective bias could prevent onset of or improve negative mood (or both), and is indeed the foundation of successful cognitive behavioural therapy [3]. Attention bias modification (ABM) is a class of training paradigms that attempt to automate this by shifting attention away from negative information (and towards positive or neutral stimuli) using cognitive tasks [4,13].

Dot-probe tasks are arguably the most common *test* of attention bias [1,14,15]. They typically involve quick presentations of two emotionally valenced stimuli, one benign (e.g. a happy or calm face/word), and the other negative (sad or angry face/word). A visual probe—often dots—replaces one item, and participants answer probe-related questions (e.g. 'Where are the dots?', 'left' or 'right'). Quicker responses to probes replacing negative than benign items suggest negatively biased attention [14,16]. This therefore assesses attention distribution in spatial orienting to emotionally salient stimuli. Dot-probe negative-avoidance ABM involves probes replacing the non-negative item 80–100% of the time [4,15,17]. Thus individuals are expected to implicitly associate the benign item and goal response over practice, and subsequently habitually bias attention to benign items. Dot-probe ABM has been associated with decreased attention bias [18] and negative affect (NA) [19–21], including reduced subjective anxiety after a gamified smartphone application-based training [21]. However, there are some reproducibility concerns [22–26], including for smartphone-based ABM [27], and the effects have not been tested at scale.

Visual-search tasks also assess affect-biased attention. Typical procedures involve positive-search trials—identifying the positive face in an array of negative face distractors, and negative-search trials—selecting the negative face surrounded by positive expressions [4,28]. Negative affective bias is reflected in quicker performances on negative- than positive-search trials, and may reflect poorer inhibition of target-irrelevant negative faces and poorer shifting of attention away from these distractors [4]. Visual-search ABM training usually consists of positive-search trials [29]. It has been associated with anxiolytic effects when delivered in the laboratory [30], clinic [31], at home [32,33] and work environments [29]. However, ABM was not found to influence bias or mood in dysphoric participants after one session [34]. A modified laboratory-based version in which moving faces must be selected before they reach the bottom of a computer screen was not found to influence bias or mood in healthy participants after either one or five laboratory sessions [35]. Moreover, these approaches have not been tested at scale.

In this study, we therefore aimed to test the efficacy of these interventions at scale, and outside the clinic ($N = 153\,385$) using a smartphone app. In total, 85% of the British public is estimated to own or have access to a smartphone in 2017 [36]. Smartphone application-based interventions provide instant access and have a relatively low barrier to entry. They could have particularly high practical utility for patients who live remotely, do not want to try or do not respond to conventional treatments. Moreover, ABM might be a novel means of delivering secondary prevention of low mood [20,37,38].

The degree to which smartphone-based ABM tasks may be effective at improving mood across the general public remains unknown; no published study has investigated the effects on a large scale. We therefore delivered dot-probe and visual-search tasks to a large sample through an online smartphone application Peak (OSF pre-registration: https://osf.io/brpxu/). The primary hypothesis was that receiving dot-probe or visual-search ABM would result in improved mood compared to sham conditions. Secondly, it was predicted that modification training would decrease negative affective bias relative to sham training on each task.

# 2. Methods

## 2.1. Participants

This study was approved by the UCL Research Ethics Committee (6199/001). Informed consent was obtained from participants through the smartphone application (www.peak.net). Participants were 153 385 self-referring application users between April 2017 and September 2018. Our aim was to test effects at scale in a convenience sample, so no power calculation was performed; we recruited as many participants as possible. Participants were randomly assigned into bias modification (active) or control (sham) training on a dot-probe (henceforth *DOT*) or visual-search (*VS*) task, or they performed no training. This resulted in DOT-active, DOT-sham, VS-active, VS-sham and no-training groups. Data were anonymized prior to analysis so no demographic information was available. Participants provided consent via the app. To participate they had to navigate to a special 'experimental' section of the app where they could participate in scientific studies. When they started the game within this

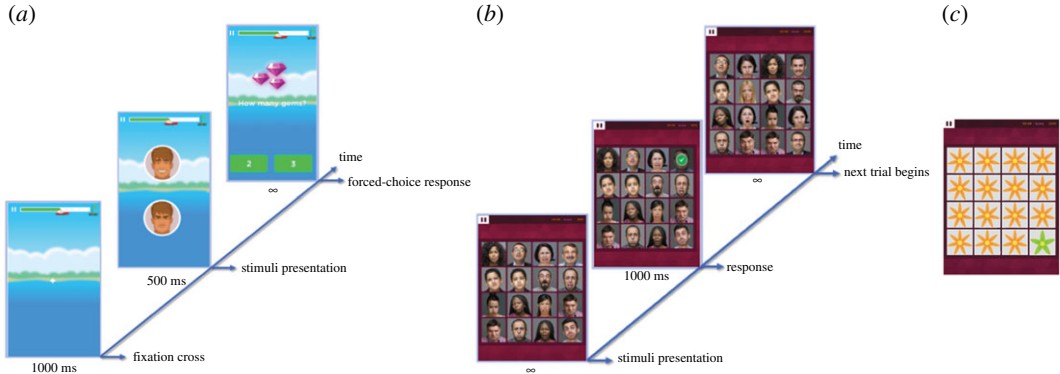

**Figure 1.** Stimuli used in DOT and VS tasks. (*a*) Task schematic for the DOT test. A fixation cross is displayed for 1000 ms, followed by a positive and negative face which disappear after 500 ms. Participants make quick responses to a question about a probe that replaces one of the faces. (*b*) Timeline of the positive-search block of the VS test. A 4 × 4 grid of faces is displayed, and participants pick the only happy face surrounded by negative faces as quickly as possible. There is a 1000 ms gap between trials where feedback is provided in the form of a green tick or red cross. (*c*) Example of VS-sham training stimuli including a five-petalled flower goal and a seven-petalled flower distractor.

section they were informed that their assigned condition may or may not train attention towards positive stimuli, and would only be informed about the condition at the end of the study. After reading the information sheet, they had to provide explicit consent. This was completely separate from the boilerplate app terms and conditions.

## 2.2. Tasks and procedure

Sessions started with mood ratings, followed by *test* versions of DOT and then VS tasks. They ended with DOT-active, DOT-sham, VS-active or VS-sham *training* dependent on assigned training version for the experiment (the subset of participants not completing any training comprise those who quit this section of the app before this final stage, some of whom later provided more mood ratings and performed repeated tests). Participants were asked to perform one session a day for at least 15 days.

## 2.3. Mood ratings

Participants completed the Positive and Negative Affect Schedule (PANAS; [39]) on the first session and after 14 consecutive days. If participants missed one or more sessions, they provided PANAS ratings again during their next session. The PANAS consists of a 10-item positive affect (PA) scale and 10-item negative affect scale. The PA questions comprised: Interested, Excited, Strong, Enthusiastic, Proud, Alert, Inspired, Determined, Attentive and Active. By contrast, the NA scale consists of Distressed, Upset, Guilty, Scared, Hostile, Irritable, Ashamed, Nervous, Jittery and Afraid. Responses were provided on a five-point scale: 1 corresponded to Not At All, 2 to A Little, 3 to Moderately, 4 to Quite A Bit and 5 to Extremely. Compound PA and NA scores were calculated by adding ratings on respective items, where values range between 10 and 50 and a higher value corresponds to feeling more positive or negative, respectively. The original Watson *et al.* [39] research found that PANAS shows high reliability (PA subscale: 0.88 internal consistency, 0.68 test–retest reliability; NA subscale: 0.87 internal consistency, test–retest reliability of 0.71). This is supported in a more recent review showing that the PANAS has high internal consistency and discriminant validity across languages, cultures and in online studies [40]. Between PANAS ratings, participants were asked to provide rating for three mood items, Happy, Anxious and Sad.

## 2.4. DOT task

### 2.4.1. Test

The DOT test started with viewing a fixation-cross in the centre of the screen for 1000 ms (figure 1*a*). Participants then viewed two vertically aligned cartoon faces, one positive and the other negative. Face positions were counterbalanced randomly such that each was on top 50% of the time. Stimuli

were removed after 500 ms, and a probe image displaying varying numbers of gems replaced one face; it was equally likely to replace each stimulus. Participants then answered a simple two-option probe-related question by clicking the corresponding answer (e.g. 'How many gems?', '2' or '3'). They had unlimited time to respond but were encouraged to act as quickly and accurately as possible. Each correct response advanced a progress meter at the top of the screen by 1 point, with visual feedback of accuracy provided by a tick or cross. The test ended once 50 trials were answered correctly (approx. 2 min). The first test was preceded by six practice trials, where the probe replaced the positive stimulus on three trials. This was followed by feedback and repeated until performed correctly.

### 2.4.2. Training

DOT-sham training followed the same procedure as the test. However, during active training the probe always replaced the positive image; this was intended to draw attention towards positive and away from negative stimuli. Participants had to complete 100 trials accurately. Incorrect trials were repeated until a correct response was provided (lasting 4–5 min).

## 2.5. VS task

### 2.5.1. Test

The VS test consisted of positive-search and negative-search blocks with 25 trials each, lasting 4–5 min (figure 1b). The order of blocks was counterbalanced across participants and sessions. In the positive-search block, participants viewed a $4 \times 4$ array of emotional faces, one of which was happy and the others negative (angry, fearful and sad—five of each). Face locations were randomized on each trial. Participants selected the happy face and were encouraged to respond as quickly as possible. Trials were separated by a 1000 ms interval where visual feedback by a green tick or red cross appeared above the selected stimulus. Trials answered incorrectly were repeated at the end of the block until answered correctly. The negative-search block involved selecting the only negative face in a grid of happy faces. 1–10 points were awarded based on reaction times for correct responses. The number of points gained was presented between blocks, and the final score after test. The first test was preceded by six practice trials (three of each positive- and negative-search), followed by feedback. Trials with erroneous responses were repeated until answered correctly.

### 2.5.2. Training

Training followed the test schematic, but consisted of four blocks with 25 trials each and varied in array type. Active training consisted of positive-search trials only with different face stimuli from test, lasting 8–10 min. Sham training involved selecting the only five-petalled flower in a $4 \times 4$ array of seven-petalled drawings of flowers (figure 1c), lasting 4–5 min.

## 2.6. Data analysis

The outcome of interest was balance of mood (henceforth *mood*) measured by the difference between total PA and total NA ratings (mood = PA − NA); a positive value represents higher positive than NA.

Unfortunately indicators of negative affective bias on tasks could not be calculated due to an oversight in the task code (outputs were collapsed across trials so we could not distinguish performance by specific trial type). Therefore, the analysis cannot assess bias and we are unable to test our second prediction (although it should be noted that training was associated with improved overall accuracy and reduced RTs collapsed across trials; all $p < 0.001$).

Univariate and mixed ANOVAs were conducted to investigate the effects of training version and time on mood. All analyses employ a significance threshold of $p < 0.05$. When Mauchly's test was significant, sphericity could not be assumed and the Greenhouse–Geisser correction was applied. We first analysed the largest group possible; those with at least two PANAS reports; to determine the effect of training over time. Subsequently, we analysed the smaller sample with at least six PANAS reports in an attempt to plot trajectories of mood change.

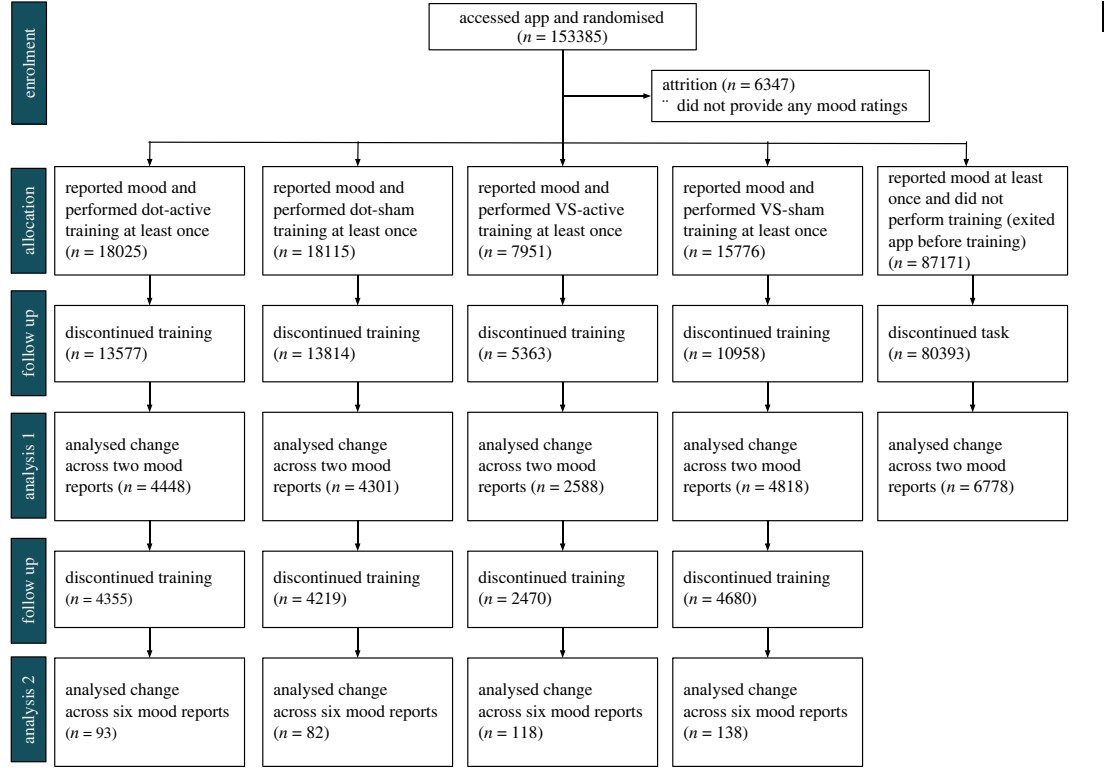

**Figure 2.** CONSORT Diagram displaying flow of participants.

# 3. Results

The flow of participants through the study is displayed in figure 2. Of the initial 153 385, 147 038 participants provided mood ratings on at least one occasion. In total, 22 933 completed at least two PANAS reports, and 471 at least six reports.

## 3.1. Change in mood over two PANAS reports

A univariate ANOVA found that the effect of training version on mood at baseline was not significant ($F_{4,22928} < 2.5$, $p > 0.05$), indicating that baseline effects were not present.

For participants who reported PANAS scores twice, a mixed ANOVA (time(1,2) × training version(DOT-active, DOT-sham, VS-active, VS-sham, no-training)) on mood found that the overall effect of time was not significant ($F_{1,22928} < 0.5$, $p > 0.1$). The effect of training version was significant ($F_{4,22928} = 8.061$, $p < 0.001$, partial $\eta^2 = 0.001$), indicating group differences in overall mood levels. However, this was qualified by a significant interaction between time and training version ($F_{4,22928} = 16.266$, $p < 0.001$, partial $\eta^2 = 0.003$; figure 3), showing that the extent of mood change differed across participants. The no-training group reduced mood more than all training groups (DOT-active: $t_{11224} = -5.293$, $p < 0.001$, $d = 0.102$, mean difference $= -1.020$, s.e. $= 0.193$; DOT-sham: $t_{11077} = -6.136$, $p < 0.001$, $d = 0.120$, mean difference $= -1.190$, s.e. $= 0.194$; VS-active: $t_{9364} = -6.07$, $p < 0.001$, $d = 0.142$, mean difference $= -1.381$, s.e. $= 0.227$; and VS-sham: $t_{11594} = -5.474$, $p < 0.001$, $d = 0.102$, mean difference $= -1.020$, s.e. $= 0.186$). No other significant differences were found between groups (all $t < 2$, $p > 0.05$). Critically, paired samples $t$-tests revealed that VS-active participants significantly increased mood from baseline ($t_{2587} = 2.476$, $p = 0.013$, $d = 0.049$, mean difference $= 0.458$, s.e. $= 0.185$) while no-training participants significantly decreased mood ($t_{6777} = -7.604$, $p < 0.001$, $d = 0.093$, mean difference $= -0.923$, s.e. $= 0.121$). No other groups significantly changed mood over time (all $t < 2$, $p > 0.05$).

## 3.2. Change in mood over six PANAS reports

When investigating mood changes in training participants providing six PANAS reports (figure 4), a mixed ANOVA (time(1,2,3,4,5,6) × training version (DOT-active, DOT-sham, VS-active, VS-sham)) on mood found that the effect of training version was not significant ($F_{3,427} < 0.5$, $p > 0.1$). The effect of

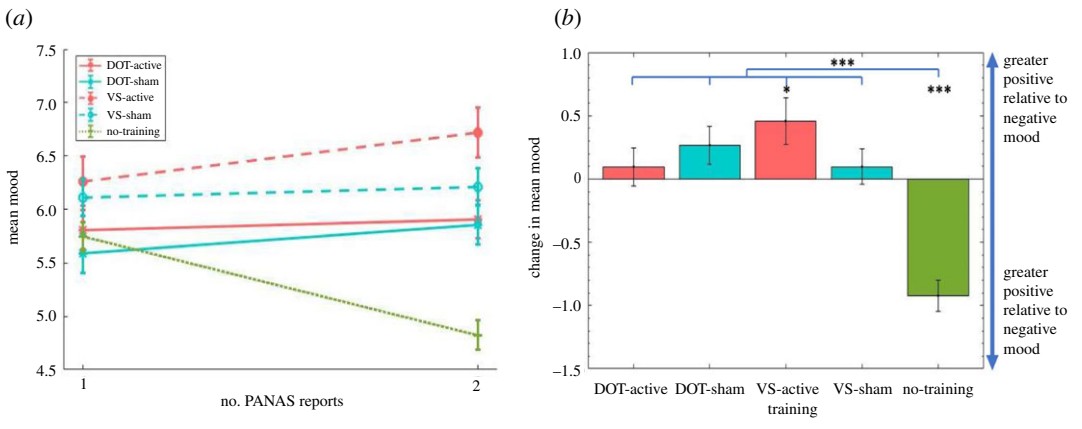

**Figure 3.** Mood by training version across two PANAS reports. (*a*) Mean mood for each group on first and second PANAS reports. (*b*) Change in mean mood across both PANAS reports for each group (same data as (*a*) but plotted more clearly). Error bars represent standard error of the mean in all figures. Single asterisks (\*) represent significant differences at the level of $p \leq 0.05$; triple asterisks (\*\*\*) represent significant differences at the level of $p \leq 0.001$.

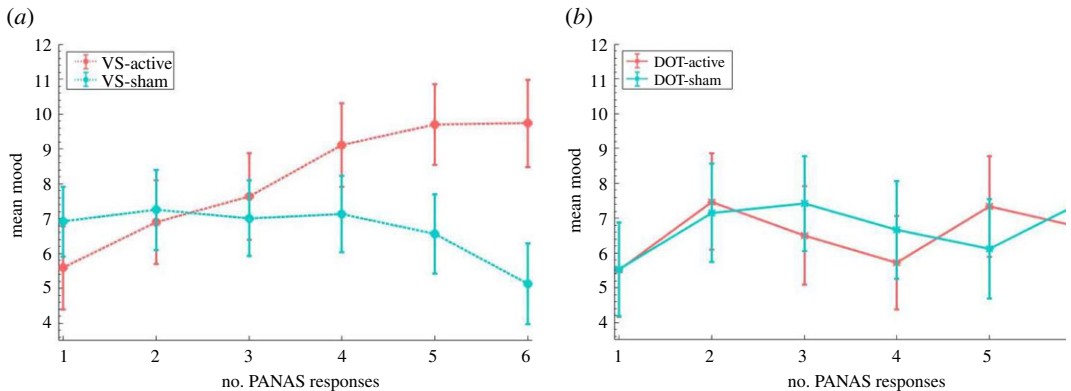

**Figure 4.** Mood by training version across six PANAS reports. (*a*) Mean mood for each VS group on the six PANAS reports. (*b*) Mean mood for each DOT group on the six PANAS reports.

time was significant ($F_{4.438,1894.973} = 2.347$, $p = 0.046$, partial $\eta^2 = 0.005$), qualified by a significant interaction between time and training version ($F_{13.314,1894.973} = 2.450$, $p = 0.002$, partial $\eta^2 = 0.017$). VS-active participants improved mood to a greater extent than VS-sham participants ($t_{254} = 3.794$, $p < 0.001$, $d = 0.474$, mean difference = 5.927, s.e. = 1.562). The DOT-sham group improved mood more than VS-sham participants ($t_{218} = 2.277$, $p = 0.024$, $d = 0.319$, mean difference = 3.722, s.e. = 1.634). No other significant differences were found (all $t < 2$, $p > 0.05$).

Post hoc paired samples *t*-tests found that VS-active participants significantly increased mood from time 1–6 ($t_{117} = 3.428$, $p = 0.001$, $d = 0.316$, mean difference = 4.144, s.e. = 1.209); no other groups significantly changed mood across six reports (all $t < 2$, $p > 0.05$). Post hoc comparisons were conducted to investigate the timepoint from which mood levels significantly differed between VS groups. Between-group *t*-tests showed that VS-active participants had significantly higher mood than sham counterparts from time 5 onwards (time 5: $n_{active} = 205$, $n_{sham} = 248$, $t_{451} = 2.174$, $p = 0.030$, $d = 0.205$, mean difference = 2.731, s.e. = 1.256; time 6: $t_{254} = 2.684$, $p = 0.008$, $d = 0.337$, mean difference = 4.598, s.e. = 1.714), but not times 1–4 (all $t < 2$, $p > 0.05$).

### 3.2.1. Exploratory analysis: change in mood based on start mood

As an exploratory analysis, we examined balance of mood over six PANAS reports for participants starting with negative (<0) versus positive (0+) baseline mood, split by task-type (VS or DOT). For DOT-training participants, a mixed ANOVA (time(1,2,3,4,5,6) × training version (DOT-active $n = 93$, DOT-sham $n = 82$) × baseline mood (positive, negative)) on mood found that the effect of training version was not significant ($F_{1,171} = 0.425$, $p > 0.1$). The effects of baseline mood ($F_{1,171} = 115.818$, $p < 0.001$, partial $\eta^2 = 0.404$) and time

($F_{4.614,789.042} = 4.439$, $p = 0.001$, partial $\eta^2 = 0.025$) were significant; there was a significant interaction between time and baseline mood ($F_{4.614,789.042} = 6.544$, $p < 0.001$, partial $\eta^2 = 0.037$). No other interactions were significant (all $p > 0.1$).

For versus-training participants, the effect of training version was not significant ($F_{1,252} = 3.432$, $p = 0.065$). The effects of baseline mood ($F_{1,252} = 114.435$, $p < 0.001$, partial $\eta^2 = 0.312$) and time ($F_{4.230,1065.835} = 7.314$, $p < 0.001$, partial $\eta^2 = 0.028$) were significant. Significant interactions were found between time and training version ($F_{4.230,1065.835} = 5.319$, $p < 0.001$, partial $\eta^2 = 0.021$), and time and baseline mood ($F_{4.230,1065.835} = 12.308$, $p < 0.001$, partial $\eta^2 = 0.047$). Interactions between training version and baseline mood ($F_{1,252} = 0.036$, $p > 0.1$), and between training version, baseline mood and time ($F_{4.230,1065.835} = 0.845$, $p > 0.1$) were not significant.

Post hoc paired samples $t$-tests were performed to further investigate changes in mood between the first and sixth PANAS reports for VS-active participants. Participants with negative baseline mood reported significantly higher mood at time 6 ($t_{36} = 5.058$, $p < 0.001$, $d = 0.832$, mean difference = 11.892, s.e. = 2.351). No significant difference was found from time 1–6 for participants with positive baseline mood ($t < 1$, $p > 0.1$) (figures 5 and 6).

### 3.2.2. Exploratory analysis: baseline mood of VS participants based on persistence with training

Finally, we investigated whether there were baseline differences between VS participants who persist with training until providing six PANAS reports compared to those who drop out before this stage (figure 7). We performed a mixed ANOVA (VS-training version (VS-active, VS-sham)×total number of PANAS reports (fewer than six reports, at least six reports) on baseline mood. The effects of VS-training version ($F_{1,23723} < 0.5$, $p > 0.1$), total number of PANAS reports ($F_{1,23723} < 0.5$, $p > 0.1$), and the interaction between VS-training version and total number of PANAS reports ($F_{1,23723} < 1.5$, $p > 0.1$) were not significant.

## 4. Discussion

To our knowledge, this is the largest observational study investigating dot-probe and visual-search ABM delivered via smartphones. There was a large amount of participant attrition, however, results from the remaining smaller sample supported the prediction that visual-search, but not dot-probe, modification can improve mood.

### 4.1. Visual-search training

The primary finding is that visual-search modification was associated with a significant, albeit small, within-group increase in positive relative to negative mood, which was not seen for the sham version. Consistent with previous investigations [27,29–31], we provide evidence that VS ABM has the potential to improve mood, even when administered at home [32,33].

While VS ABM appears to have some effect, we could not determine what exactly drove mood improvements due to inability to calculate changes in bias. Cognitive models suggest this may be related to decreasing negative attention bias by improving the ability to positively direct attention while inhibiting processing of negative distractors [4,29]. This would be evidenced by VS-active participants improving positive-search speed to a greater extent than negative-search speed.

Further research is required to determine if VS ABM is effective generally or for a specific subset of the population. The exploratory analysis accounting for positive or negative baseline mood indicates that the mood improvement associated with VS-active training is driven by participants with low starting mood. Those with negative mood before training improved mood over time (with a large effect size); those with positive baseline mood did not reduce mood over time. While the sample was unselected, this suggests ABM may be most effective for participants who start with low mood. VS ABM is expected to improve highly anxious affect [30–33], but previous research does not have strong implications for depressed or non-negative baseline mood. Nevertheless, it appears that training can be effective, albeit with a small effect size, in a naturalistic self-selected small sample of smartphone users who persist with training over time.

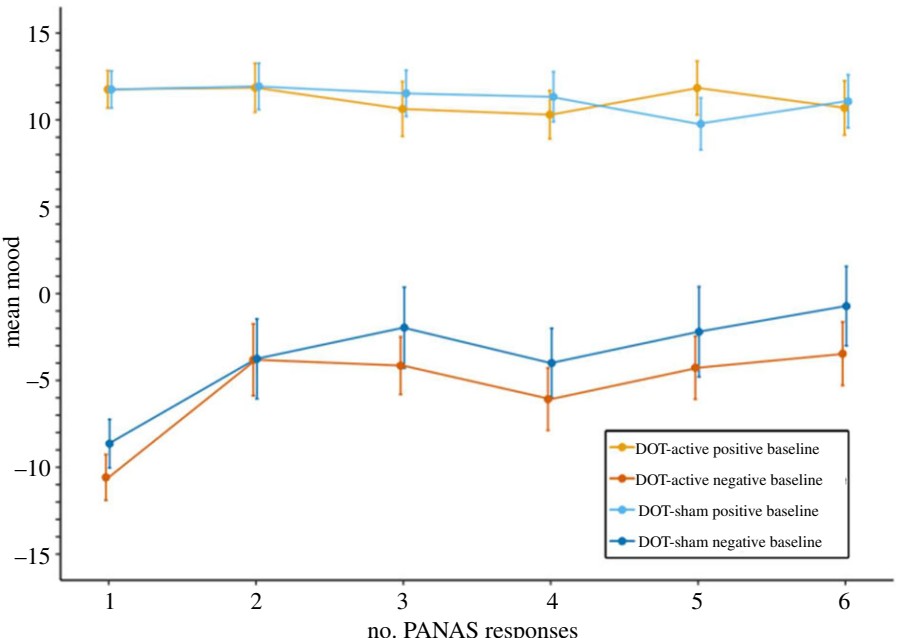

**Figure 5.** Mood by training version for DOT participants across six PANAS reports, split by baseline mood (positive versus negative).

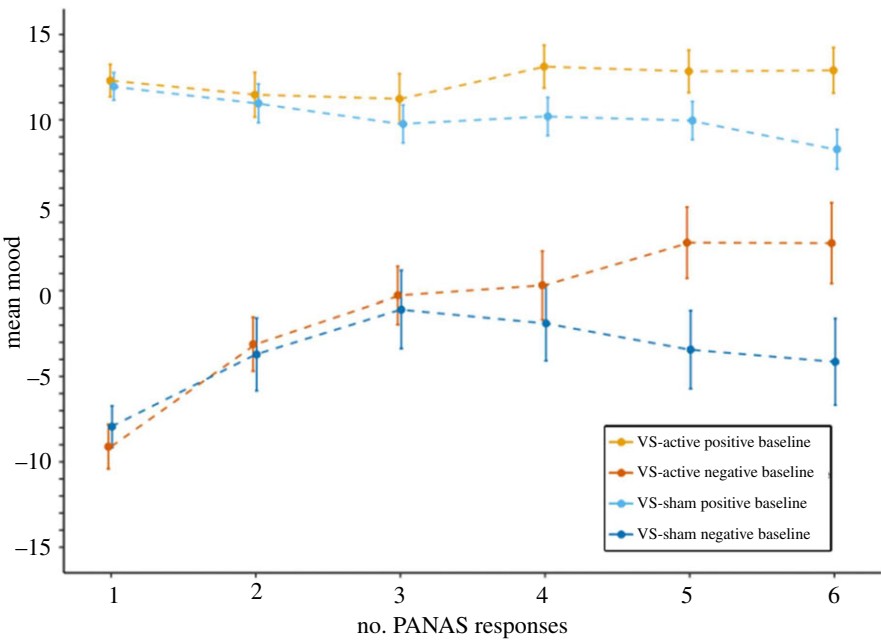

**Figure 6.** Mood by training version for VS participants across six PANAS reports, split by baseline mood (positive versus negative).

## 4.2. Dot-probe training

The presented dot-probe intervention was not effective as the DOT-active group did not significantly change mood following training.

One possibility is that dot-probe ABM is simply not effective. Several investigations failed to find that it improves heightened or severely low mood [22–25]; some have questioned its rationale since negative-avoidance may not reflect 'normal' attention processing and negative-attendance may not characterize low mood [22,26,41]. However several studies associated dot-probe ABM with mood improvements in clinical or elevated depression and anxiety [18–20], where a moderation analysis by Linetzky et al. [18] showed a significant effect of dot-probe ABM on self-reported levels of anxiety when administered in

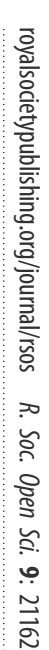

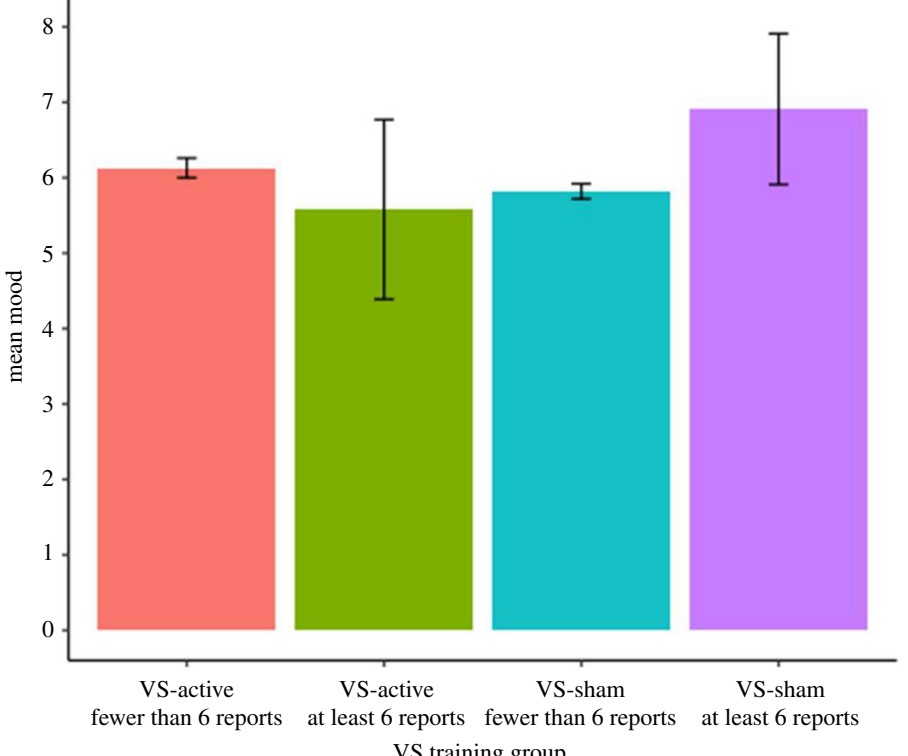

**Figure 7.** Baseline mood for VS participants, grouped by the total number of PANAS reports subsequently completed.

clinics ($d = 0.34$) but not at home ($d = -0.10$). Therefore, it may be only effective on severely low mood, and more impactful when delivered in clinics than at home.

Our DOT ABM procedure may have also been ineffective due to task characteristics. One study reported that dot-probe modification only effectively changed bias when presenting prototypical and intense fear expressions [42]. Our face drawings may have been too simple, or not salient enough. Another possibility is that stimulus presentation timings were sub-optimal. Future investigations could use an app to present real faces with intense expressions, decrease probe visibility (present more/smaller/blurred probes) and/or modify stimulus timings.

### 4.3. Limitations and future directions

A key limitation is that predictions about effects of ABM on bias could not be tested. Therefore, we are unable to support or falsify our second prediction that dot-probe or visual-search ABM would decrease negative affective bias relative to sham training, and explore whether mood changes were mediated by change in bias. Future research will acquire trial type information, so we will be able to test the interaction between bias and mood change, and provide insights on crossover effects of tasks.

We could not analyse whether VS ABM effectiveness is contingent on initial mood because PANAS ratings were the only indicator of baseline mood. Further research could administer multiple mood scales, as well as demographic questionnaires, at baseline to explore the impact of pre-treatment differences on subsequent PANAS changes. Similarly, follow-up PANAS ratings should be collected after app completion to identify if VS ABM has sustained benefits.

It is unclear why large numbers of participants did not perform training. The most convincing explanation is that no-training participants quit the application before training on the first session, and later returned. This group were not fully engaged so may have a selection bias that drives their large decrease in mood.

Similarly, there was asymmetric drop-off in different conditions. Fewer people completed at least one VS-active than VS-sham training session, for instance. Thus those assigned to active intervention may have been less likely to stick with training, biasing resultant effects. The analysis comparing VS-training participants based on whether they complete six PANAS report suggests that baseline

mood does not account for persistence with (or drop-off from) training. Based on the number of participants reporting mood for at least six reports, it is also possible that participants who persist with ABM find VS training more engaging than DOT training. Regardless, there appears to be a high attrition rate in participants using smartphone-delivered ABM—only 27% of training participants reporting baseline PANAS subsequently report it again. The 15-day gap between PANAS reports was chosen to mirror approaches used in many clinical trials of psychiatric treatments. However this led to loss of PANAS reports from participants who performed multiple, but less than 15, consecutive sessions as well as differences in intervals between PANAS reports across participants (since additional PANAS reports are presented following a missed session). Future investigations should fix this confound, as well as investigate why some people persist and others do not complete training.

Research would also benefit from replicating this investigation in a laboratory environment to present a more controlled setting. This would limit concerns about external factors influencing test performance, and biased drop-off. For instance, concerns about participants letting the task run unattended in the background meant that our tasks were self-paced, which likely impacted reaction times and attention to the task. Ultimately, however, this study was run in the environment in which application interventions would be theoretically used, so at a population level, any biased sampling effects would likely also be present. Thus, the findings have implications for how effective smartphone-based ABM would be 'in the wild'.

It should be noted that there are differences between the DOT and VS task designs. Both tasks were based on previously used designs, but differed in the amount and timing of when points were awarded which could explain efficacy differences across tasks. Moreover, VS-sham training involved searching an array of flowers rather than faces, based on classic designs found in the literature [29,31–33]. Fewer participants may finish the first VS-active training component because finding the anomalous face takes more time and may require more cognitive effort than finding the anomalous flower. The procedure may have inadvertently resulted in a skewed VS-active sample. For example, individuals who stick with longer or more difficult interventions may be more likely to improve from VS ABM. Conversely, performing longer or more effortful interventions may be related to increased likelihood of participants rating themselves as improved. Future research could attempt to equalize the difficulty and length of each trial across both VS-conditions, e.g. by presenting both positive-search and negative-search face trials during sham training. Relatedly, our inability to determine the cognitive training effects on bias means that we cannot rule out that the mood effects are due to these differences in the type of stimuli presented.

Finally, it should be noted that the effect size of the successful mood improvement is very small ($d = 0.05$). It cannot therefore be recommended as a substitute for conventional treatment. Nevertheless, at a population level, such small effects can have quite a profound impact. They are also low cost, easy to access and have limited potential for side-effects.

## 5. Conclusion

This study advances prior work by highlighting that smartphones can be used to successfully improve mood in a large sample through visual-search ABM. The presented dot-probe approach is not effective at changing mood, but does nevertheless act as a negative control indicating that the VS ABM mood improvement is not a generic effect of completing training sessions. This study therefore has important practical implications because smartphones allow access to training remotely, at relatively low cost and are easily accessible in many countries.

Ethics. This study was approved by the UCL Research Ethics Committee (6199/001).

Data accessibility. Code used to analyse data is stored in GitHub: https://github.com/lyshc/bias-modification and is archived within the Zenodo repository: https://doi.org/10.5281/zenodo.5593212 [43].

Authors' contributions. A.C.: formal analysis, validation, visualization, writing—original draft, writing—review and editing; O.R.: conceptualization, formal analysis, funding acquisition, investigation, methodology, project administration, resources, supervision, validation, writing—original draft, writing—review and editing.

All authors gave final approval for publication and agreed to be held accountable for the work performed therein.

Conflict of interest declaration. At the time of writing, O.R. is an Editor of Royal Society Open Science, but had no involvement in the review or assessment of the paper.

Funding. O.R. was funded by a Medical Research Council Career Development Award (grant no. MR/K024280/1).

Acknowledgements. This study was completed as part of a consultancy agreement between O.R. and Peak. O.R. and Peak developed the study, Peak hosted the tasks and the assessment and collected the data. Analysis was completed by A.C.

and O.R., both of whom wrote the paper. Peak hold a financial interest in the tasks described in this study. The authors declare no conflict of interest and committed in advance to publishing this study regardless of the direction of the findings. We would like to thank Peak for providing the app for research purposes.

Disclosures. Miss Chelliah reports no financial relationships with commercial interests.

Dr Robinson reports no financial relationships with commercial interests.

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
