## [Peer Review File · Royal Society Open Science]

Review History

Decision letter (RSOS-211242.R0)

Dear Miss Chelliah

The Editors assigned to your paper RSOS-211242 "Efficacy of attention bias modification via smartphones in a large population sample" have made a decision based on their reading of the paper.

Regrettably, based on a thorough assessment by the Associate Editor, the manuscript has been rejected in its current form. However, a new manuscript may be submitted which takes into consideration these comments.

We invite you to respond to the comments supplied below and prepare a resubmission of your manuscript. Below the Editors' comments (where applicable) we provide additional requirements. We provide guidance below to help you prepare your revision.

Please note that resubmitting your manuscript does not guarantee eventual acceptance, and we do not generally allow multiple rounds of revision and resubmission, so we urge you to make every effort to fully address all of the comments at this stage.

Please resubmit your revised manuscript and required files (see below) no later than 14-Feb-2022. Note: the ScholarOne system will 'lock' if resubmission is attempted on or after this deadline. If you do not think you will be able to meet this deadline, please contact the editorial office immediately.

Thank you for submitting your manuscript to Royal Society Open Science and we look forward to receiving your resubmission. If you have any questions at all, please do not hesitate to get in touch.

on behalf of Dr Gina Grimshaw (Associate Editor) and Essi Viding (Subject Editor)
openscience@royalsociety.org

Associate Editor Comments to Author (Dr Gina Grimshaw):
Associate Editor

Comments to the Author:

This is a "real world" experimental study on the effects of two smart-phone apps that provide attentional bias modification (ABM) using either dot-probe (DOT) or visual search (VS) methods. Participants were randomly assigned to either an active or sham training condition for one of the tasks. A group who started the app but didn't proceed to training served as a control group. Participants were intended to use the app for 15 consecutive days. It is not clear how many did so, but only a small proportion of the sample persisted for 6 days, providing the primary test of hypotheses. Although I appreciate that assessment of such interventions "in the wild" are particularly valuable, especially when conducted so transparently, I also have concerns with how the findings are currently described and interpreted. Given that scientific endorsement of a commercial product is likely to be influential (including financially), I think it is important to address these concerns now before considering peer review and possible publication. I will enumerate my concerns here:

1. The primary concern is the high rate of attrition - of the approximately 147,000 people who start the app (i.e. provide mood ratings), only 22,000 participate a second day (providing a one-day follow-up), and only 471 participate 6 times (n's of 82-138 per condition), providing the primary test (and support for) the hypothesis. This rate of attrition has several implications. First, the authors should not really claim that this is a "large" population sample, given usable n's of around 100 per condition. Secondly, an app for which only 0.3% of participants continue use for 6 days (of the 15 intended) is perhaps not all that successful, even if it does improve mood for some of those participants. I think the authors should therefore be more conservative in describing the efficacy of the VS app. Scientifically, such attrition also compromises internal validity. What do we know about the people who drop out of the study? Given that most of them provided mood ratings on Day 1, it would be useful to compare the drop-outs across conditions to ensure that differential drop-out doesn't account for the increase in mood with VS training. This is particularly important given the decrease in mood in the "no-training" group.
2. The authors indicate that the study was approved by their institutional ethics committee and that participants gave informed consent. Given that the study involves both active and sham

training conditions, I would like more detail on specifically what participants were told, and how they were informed. For example, did they acknowledge their consent to participant in blinded study in which they might or might not be assigned a functional version of the training? Was consent separate from the overall boilerplate user agreement that we know people don't read? If deception was involved, was there any way to debrief participants on it (especially given the high drop-out rate)?

3. The technical failure to capture data necessary to measure attentional bias is really unfortunate (I can only imagine how the authors must have felt to discover it). Unfortunate though it is, it also really limits the scientific contribution of the study, as we cannot know whether there was any attentional bias to be modified, or whether training modified it. Without the ability to probe the mechanisms by which mood may have improved in the VS-active condition, it is difficult to attribute the effects to attentional bias modification. This is particularly the case with the VS task, in which the sham condition involves non-face stimuli; thus effects could be due to the use of social (faces) versus non-social stimuli, and not attention.

Reviewer comments to Author:

===PREPARING YOUR MANUSCRIPT===

===PREPARING YOUR REVISION IN SCHOLARONE===

To revise your manuscript, log into <https://mc.manuscriptcentral.com/rsos> and enter your Author Centre - this may be accessed by clicking on "Author" in the dark toolbar at the top of the

page (just below the journal name). You will find your manuscript listed under "Manuscripts with Decisions". Under "Actions", click on "Create a Revision".

Author's Response to Decision Letter for (RSOS-211242.R0)

See Appendix A.

RSOS-211629.R0

Review form: Reviewer 1 (Thalia Eley)

Is the manuscript scientifically sound in its present form?

Yes

Are the interpretations and conclusions justified by the results?

Yes

Is the language acceptable?

Yes

Do you have any ethical concerns with this paper?

No

Have you any concerns about statistical analyses in this paper?

No

Recommendation?

Accept with minor revision (please list in comments)

Comments to the Author(s)

RSOS-211629

This study examines the degree to which smartphone delivered ABM tasks improve mood in a large volunteer sample. This is a very important and timely question given high rates of anxiety in the population, reluctance with regard to treatment seeking (especially in some group) and long wait lists in many areas for psychological treatment.

Abstract

1. I agree with the prior feedback that the Ns for the various analyses need to be clearer. Whilst the edit made is helpful, in my view, the first sentence of the results needs to give the actual (range of) Ns for the results being reported.

Methods

2. It is a shame the data were anonymised such that no demographic data was available at all. Something to think about designing differently another time.

3. Please provide the psychometric properties of the PANAS. These are important given your repeated assessment design.

Results

4. Figure 3b could be clearer. It took me quite a while to realise that you were comparing ALL training groups (regardless of being active or sham) with the non-training group. I realise you do say this in the text on page 11, but that paragraph is also rather dense and anyway it is important that all information in a figure can be understood without recourse to the text. It would have helped me if either the bracket above the 4 training groups had little lines down to each

group to show they are all being included, as I initially assumed it was only the outer edge two (i.e. DOT-active and VS-sham) which seemed bizarre!

5. The additional analyses presented in the response regarding the differences between those who persisted and those who did not are useful. It is good (and a little surprising) to see there were no marked differences between these groups.

Discussion

6. Is it possible that participants simply find the VS ABM more engaging/interesting to do than the DOT ABM (which is tedious for everyone surely!).

Minor point:

7. I did not find this statement (in the author contributions) clear: "Analysis was completed by AC and OJR who wrote the paper." Does this mean that AC and OJR both wrote the paper or just OJR?

In support of Open Science I sign my reviews: Thalia Eley.

Decision letter (RSOS-211629.R0)

Dear Miss Chelliah

On behalf of the Editors, we are pleased to inform you that your Manuscript RSOS-211629 "Efficacy of attention bias modification via smartphones in a large population sample" has been accepted for publication in Royal Society Open Science subject to minor revision in accordance with the referees' reports. Please find the referees' comments along with any feedback from the Editors below my signature.

Please submit your revised manuscript and required files (see below) no later than 7 days from today's (ie 28-Jan-2022) date. Note: the ScholarOne system will 'lock' if submission of the revision is attempted 7 or more days after the deadline. If you do not think you will be able to meet this deadline please contact the editorial office immediately.

Kind regards,
Royal Society Open Science Editorial Office

on behalf of Dr Gina Grimshaw (Associate Editor) and Essi Viding (Subject Editor)
openscience@royalsociety.org

Associate Editor Comments to Author (Dr Gina Grimshaw):

Associate Editor

Comments to the Author:

Dear Dr Chelliah,

Thank you for submitting your revised manuscript "Efficacy of attention bias modification via smartphones in a large population study" to Royal Society Open Science. I think you have addressed the concerns in my original decision letter well. I was also able to secure one external review (from Thalia Eley, who has signed their review). The reviewer has made some recommendations that I think will improve the manuscript. I am happy to recommend acceptance of the manuscript pending these minor revisions.

Kind regards

Gina Grimshaw

Associate Editor

Reviewer comments to Author:

Reviewer: 1

Comments to the Author(s)

RSOS-211629

This study examines the degree to which smartphone delivered ABM tasks improve mood in a large volunteer sample. This is a very important and timely question given high rates of anxiety in the population, reluctance with regard to treatment seeking (especially in some group) and long wait lists in many areas for psychological treatment.

Abstract

1. I agree with the prior feedback that the Ns for the various analyses need to be clearer. Whilst the edit made is helpful, in my view, the first sentence of the results needs to give the actual (range of) Ns for the results being reported.

Methods

2. It is a shame the data were anonymised such that no demographic data was available at all. Something to think about designing differently another time.

3. Please provide the psychometric properties of the PANAS. These are important given your repeated assessment design.

Results

4. Figure 3b could be clearer. It took me quite a while to realise that you were comparing ALL training groups (regardless of being active or sham) with the non-training group. I realise you do say this in the text on page 11, but that paragraph is also rather dense and anyway it is important that all information in a figure can be understood without recourse to the text. It would have helped me if either the bracket above the 4 training groups had little lines down to each group to show they are all being included, as I initially assumed it was only the outer edge two (i.e. DOT-active and VS-sham) which seemed bizarre!

5. The additional analyses presented in the response regarding the differences between those who persisted and those who did not are useful. It is good (and a little surprising) to see there were no marked differences between these groups.

Discussion

6. Is it possible that participants simply find the VS ABM more engaging/interesting to do than the DOT ABM (which is tedious for everyone surely!).

Minor point:

7. I did not find this statement (in the author contributions) clear: "Analysis was completed by AC and OJR who wrote the paper." Does this mean that AC and OJR both wrote the paper or just OJR?

In support of Open Science I sign my reviews: Thalia Eley.

===PREPARING YOUR MANUSCRIPT===

one version should clearly identify all the changes that have been made (for instance, in coloured highlight, in bold text, or tracked changes);

===PREPARING YOUR REVISION IN SCHOLARONE===

-- If you are requesting an article processing charge waiver, you must select the relevant waiver option (if requesting a discretionary waiver, the form should have been uploaded, see 'File upload' above).

-- If you have uploaded any electronic supplementary (ESM) files, please ensure you follow the guidance at <https://royalsociety.org/journals/authors/author-guidelines/#supplementary-material> to include a suitable title and informative caption. An example of appropriate titling and captioning may be found at https://figshare.com/articles/Table_S2_from_Is_there_a_trade-off_between_peak_performance_and_performance_breadth_across_temperatures_for_aerobic_scope_in_teleost_fishes_/3843624.

Author's Response to Decision Letter for (RSOS-211629.R0)

See Appendix B.

Decision letter (RSOS-211629.R1)

Dear Miss Chelliah:

I am pleased to inform you that your manuscript entitled "Efficacy of attention bias modification via smartphones in a large population sample" is now accepted for publication in Royal Society Open Science.

Please remember to make any data sets or code libraries 'live' prior to publication, and update any links as needed when you receive a proof to check - for instance, from a private 'for review' URL to a publicly accessible 'for publication' URL. It is also good practice to add data sets, code and other digital materials to your reference list.

Royal Society Open Science is a fully open access journal. A payment may be due before your article is published. Our partner Copyright Clearance Center's RightsLink for Scientific Communications will contact the corresponding author about your open access options from the email domain @copyright.com (if you have any queries regarding fees, please see <https://royalsocietypublishing.org/rsos/charges> or contact authorfees@royalsociety.org).

on behalf of Dr Gina Grimshaw (Associate Editor) and Dr Essi Viding (Subject Editor).

Follow Royal Society Publishing on Twitter: @RSocPublishing
Follow Royal Society Publishing on Facebook:
<https://www.facebook.com/RoyalSocietyPublishing/>

Read Royal Society Publishing's blog:
<https://royalsociety.org/blog/blogsearchpage/?category=Publishing>

Appendix A

Response to Comments by the Editor

Dear Dr Gina Grimshaw,

Thank you very much for your helpful comments and the opportunity to revise and resubmit our manuscript. We believe that addressing these suggestions has improved the manuscript substantially.

Below we present comments in italics and provide our responses. The edited sections of the manuscript are highlighted in yellow, with comments added where relevant.

We hope that you find the manuscript now suitable for review.

Best wishes,

Alysha Chelliah & Oliver Robinson

Editorial comments

This is a "real world" experimental study on the effects of two smart-phone apps that provide attentional bias modification (ABM) using either dot-probe (DOT) or visual search (VS) methods. Participants were randomly assigned to either an active or sham training condition for one of the tasks. A group who started the app but didn't proceed to training served as a control group. Participants were intended to use the app for 15 consecutive days. It is not clear how many did so, but only a small proportion of the sample persisted for 6 days, providing the primary test of hypotheses. Although I appreciate that assessment of such interventions "in the wild" are particularly valuable, especially when conducted so transparently, I also have concerns with how the findings are currently described and interpreted. Given that scientific endorsement of a commercial product is likely to be influential (including financially), I think it is important to address these concerns now before considering peer review and possible publication. I will enumerate my concerns here:

1. The primary concern is the high rate of attrition - of the approximately 147,000 people who start the app (i.e. provide mood ratings), only 22,000 participate a second day (providing a one-day follow-up), and only 471 participate 6 times (n's of 82-138 per condition), providing the primary test (and support for) the hypothesis. This rate of attrition has several implications. First, the authors should not really claim that this is a "large" population sample, given usable n's of around 100 per condition.

Thank you for the evaluation. The comments on the rate of attrition are fair points. While our primary focus and test of hypotheses centred on comparing mood across two reports in what we consider a large sample (22,933 participants), we agree that the analysis of six reports is not based on a large sample (~100 participants per condition). This latter analysis was performed to investigate trajectories of mood change over a longer time scale (as noted in the Data Analysis section of the Methods).

We have now amended phrasing – most prominently we do not claim that visual-search significantly improved mood in a “large sample” in the abstract and have amended it to read:

“There was a large amount of participant attrition. However in the remaining smaller sample, results supported...Although this effect size is very small and subject to large participant drop-off;”

At the start of the discussion we now state

“There was a large amount of participant attrition, however, results from the remaining smaller sample supported the prediction that visual-search, but not dot-probe, modification can improve mood.”

As well as

“Nevertheless it appears that training can be effective, albeit with a small effect size, in a naturalistic self-selected small sample of smartphone users who persist with training over time”.

Along with amendments outlined below, we hope that the distinction between the goal and ultimate sample sizes are now clear.

Secondly, an app for which only 0.3% of participants continue use for 6 days (of the 15 intended) is perhaps not all that successful, even if it does improve mood for some of those participants. I think the authors should therefore be more conservative in describing the efficacy of the VS app. Scientifically, such attrition also compromises internal validity. What do we know about the people who drop out of the study? Given that most of them provided mood ratings on Day 1, it would be useful to compare the drop-outs across conditions to ensure that differential drop-out doesn't account for the increase in mood with VS training. This is particularly important given the decrease in mood in the "no-training" group.

We apologise for any wording that overstates the success and efficacy of the visual-search training delivered via the app. We have now amended this (see responses above).

The comment on differential drop-out based on baseline mood is a good point. As suggested, we have analysed and included further information about VS-training patients based on their baseline mood. Specifically we have (a) included a graph of baseline mood by VS-training condition for participants who do and do not persist with training (i.e. baseline mood for VS-active and VS-sham participants who do report mood six times, alongside with baseline mood for participants who drop-off before this stage; see Figure 7).

Figure 7 Baseline mood for VS participants, grouped by the total number of PANAS reports subsequently completed

and (b) performed an ANOVA to look at baseline differences in mood (no differences were found in these groups):

Exploratory analysis: baseline mood of VS participants based on persistence with training

Finally, we investigated whether there were baseline differences between VS-participants who persist with training until providing six PANAS reports compared to those who dropout before this stage (Figure 7). We performed a mixed ANOVA (VS-training version(VS-active, VS-sham) x total number of PANAS reports(fewer than six reports, at least six reports) on baseline mood. The effects of VS-training version ($F(1,23723)<0.5,p>.1$), total number of PANAS reports ($F(1,23723)<0.5,p>.1$), and the interaction between VS-training version and total number of PANAS reports ($F(1,23723)<1.5,p>.1$) were not significant.

In short, there is no evidence for a mood confound in the different groups. Based on the findings, we adjusted our discussion:

“The analysis comparing VS-training participants based on whether they complete six PANAS report suggests that baseline mood does not account for persistence with (or drop-off from) training”

“Nevertheless it appears that training can be effective, albeit with a small effect size, in a naturalistic self-selected small sample of smartphone users who persist with training over time.”

We hope this stresses that we take a conservative reading of our findings. Alongside our comments on the limitations posed by high attrition rates in the manuscript, we hope that readers can use these findings to make balanced judgements about the efficacy of VS training, and on app usage in self-referring samples outside clinical/controlled settings.

2. The authors indicate that the study was approved by their institutional ethics committee and that participants gave informed consent. Given that the study involves both active and sham training conditions, I would like more detail on specifically what participants were told, and how they were informed. For example, did they acknowledge their consent to participant in blinded study in which they might or might not be assigned a functional version of the training? Was consent separate from the overall boilerplate user agreement that we know people don't read? If deception was involved, was there any way to debrief participants on it (especially given the high drop-out rate)?

To participate in the study, participants had to navigate to a special research/experimental section of the app and select the game. This was after they had seen, and was separate from, the boilerplate user agreement that the reviewer refers to. After selecting the game, participants were informed in a clear consent form that they may be assigned to a version of training that is either 1) intended to train attention towards positive stimuli, and result in an improved mood state, or 2) intended to train attention towards all stimuli

equally, translating in no effect on mood state. For example, the form stated that “this is experimental, so you may be assigned to a sham condition which is designed to have no effect. You will not know which group you were in until the study has finished”. Participants were also required to read an information page and respond to a consent form specific to the study on the app, before participating.

We have added a condensed version of this description to the manuscript:

“Participants provided consent via the app. To participate they had to navigate to a special ‘experimental’ section of the app where they could participate in scientific studies. When they started the game within this section they were informed that their assigned condition may or may not train attention towards positive stimuli, and would only be informed about the condition at the end of the study. After reading the information sheet, they had to provide explicit consent. This was completely separate from the boilerplate app terms and conditions.”

For completeness, since this rebuttal will form part of the open materials associated with this paper, the full text of the information sheet and consent form is provided below:

Information sheet:

“We would like to invite you to participate in this research project. You should only participate if you want to and if you are over 18 years old. Choosing not to take part will not disadvantage you in any way. Before deciding to participate you should read the following information carefully and discuss with others if you wish. If anything is not clear, or you would like more information please contact us on the details above. In this experiment you will be asked to play a short game which will appear on the screen behind emotional stimuli. You will also be asked some questions about how you are feeling. This game will take somewhere between 5-15 minutes. We hope that you will then complete the same game everyday for the next two weeks. The idea is that we are either 1) training your attention towards positive stimuli and that this will translate into an improvement in your mood state or 2) training you equally towards all stimuli and that this will translate into no effect on your mood state. Note that this is experimental, so you may be assigned to a sham condition which is designed to have no effect. You will not know which group you were in until the study has finished. There are no anticipated risks or benefits associated with participation in this study.

It is up to you whether or not to take part. If you choose not to participate, you won't incur any penalties or lose any benefits to which you might have been entitled. You can select below to have a copy of this information emailed to you for your records. After agreeing to this you will be asked to confirm your consent. Note that even after agreeing to take part, you can still withdraw at any time without giving a reason

All data will be collected and stored in accordance with the UK Data Protection Act 1998. No data that can be linked back to you personally will be shared with the researchers.

Thank you for reading this information sheet and for considering take part in this research.

email me a copy of this []

I confirm that I wish to continue []”

Consent form:

“Please complete this form after you have read the Information Sheet and/or listened to an explanation about the research.

Title of Project: Affective Bias modification via smartphones

This study has been approved by the UCL Research Ethics Committee (Project ID Number): 6199/001

Thank you for your interest in taking part in this research.

If you have any questions arising from the Information Sheet or explanation already given to you, please email science@peak.net. You can go back to the information sheet by clicking ‘go back’ below.

Please confirm the following

- I have read the Information Sheet, and understand what the study involves.
- I understand that if I decide at any time that I no longer wish to take part in this project, I can simply stop without further explanation and without penalty
- I consent to the processing of my information for the purposes of this research study.
- I understand that such information will be treated as strictly confidential and handled in accordance with the provisions of the Data Protection Act 1998.
- I agree that the research project named above has been explained to me to my satisfaction and I agree to take part in this study.
- I agree that my data, after it has been fully anonymised, can be shared with other researchers
- I am over 18 years of age

Please select if you would like a copy of this emailed to you []

I confirm that the above is correct and that I wish to continue []”

3. The technical failure to capture data necessary to measure attentional bias is really unfortunate (I can only imagine how the authors must have felt to discover it). Unfortunate though it is, it also really limits the scientific contribution of the study, as we cannot know whether there was any attentional bias to be modified, or whether training modified it. Without the ability to probe the mechanisms by which mood may have improved in the VS-active condition, it is difficult to attribute the effects to

attentional bias modification. This is particularly the case with the VS task, in which the sham condition involves non-face stimuli; thus effects could be due to the use of social (faces) versus non-social stimuli, and not attention.

This is a valid point, and we agree that the data limit the ability to probe attentional bias. We expanded our discussion point in the Limitations subheading about differences in visual-search conditions and suggest how the task could be altered in future research. Combined with the above modifications, the intention is to emphasise that the apparent improvement in VS-active training could be related to VS ABM working in participants who persist with long and more effortful training, or conversely that participants who persist are more likely to rate themselves as having improved.

“Moreover VS sham-training involved searching an array of flowers rather than faces, based on classic designs found in literature (Dandeneau et al., 2007; Waters et al. 2013, 2015, 2016). Fewer participants may finish the first VS-active training component because finding the anomalous face takes more time and may require more cognitive effort than finding the anomalous flower. The procedure may have inadvertently resulted in a skewed VS-active sample. For example, individuals who stick with longer or more difficult interventions may be more likely to improve from VS ABM. Conversely, performing longer or more effortful interventions may be related to increased likelihood of participants rating themselves as improved. Future research could attempt to equalise the difficulty and length of each trial across both VS-conditions, e.g. by presenting both positive-search and negative-search face trials during sham-training. Relatedly, our inability to determine the cognitive training effects on bias means that we cannot rule out that the mood effects are due to these differences in the type of stimuli presented.”

Appendix B

Response to Comments by the Editor

Dear Dr Gina Grimshaw,

Thank you very much for sharing these helpful comments and the opportunity to revise and publish our manuscript.

Below we present comments in italics and provide our responses to comments by Thalia Eley, our reviewer. The newly edited sections of the manuscript are highlighted in blue. Earlier amendments remain highlighted in yellow.

We hope that you find the manuscript now suitable for publication.

Best wishes,

Alysha Chelliah & Oliver Robinson

Editorial comments

Abstract

1. I agree with the prior feedback that the Ns for the various analyses need to be clearer. Whilst the edit made is helpful, in my view, the first sentence of the results needs to give the actual (range of) Ns for the results being reported.

We have added that Ns per analysis to the abstract. The modified sentence now reads:

“However in the remaining smaller sample, results supported the prediction that visual-search modification would result in improved mood (95% CI .10 to .82; $p=.01$, $d=.05$, $N=2,588$ after two mood ratings; 95% CI 1.75 to 6.54; $p=.001$, $d=.32$, $N=118$ after six mood ratings), which was not seen for the sham version ($N=4,818$ after two mood rating; $N=138$ after six mood ratings).”.

Methods

2. It is a shame the data were anonymised such that no demographic data was available at all. Something to think about designing differently another time.

We acknowledge this and include it under the *Limitations and future directions* subheading of the Discussion section, which reads:

“Further research could administer multiple mood scales, as well as demographic questionnaires, at baseline to explore the impact of pre-treatment differences on subsequent PANAS changes.”

3. *Please provide the psychometric properties of the PANAS. These are important given your repeated assessment design.*

We have added this to the *Mood ratings* subsection of the *Methods*. Which reads:

“The original Watson et al. (1988) research found that PANAS shows high reliability (PA subscale: 0.88 internal consistency, 0.68 test-retest reliability; NA subscale: 0.87 internal consistency, test-retest reliability of 0.71. This is supported in a more recent review showing that the PANAS has high internal consistency and discriminant validity across languages, cultures, and in online studies (Díaz-García et al., 2020)”

Results

4. *Figure 3b could be clearer. It took me quite a while to realise that you were comparing ALL training groups (regardless of being active or sham) with the non-training group. I realise you do say this in the text on page 11, but that paragraph is also rather dense and anyway it is important that all information in a figure can be understood without recourse to the text. It would have helped me if either the bracket above the 4 training groups had little lines down to each group to show they are all being included, as I initially assumed it was only the outer edge two (i.e. DOT-active and VS-sham) which seemed bizarre!*

As suggested, we have updated the figure to include lines down to all four groups (rather than only including the two outer lines of the bracket).

5. *The additional analyses presented in the response regarding the differences between those who persisted and those who did not are useful. It is good (and a little surprising) to see there were no marked differences between these groups.*

Thank you for this feedback.

Discussion

6. *Is it possible that participants simply find the VS ABM more engaging/interesting to do than the DOT ABM (which is tedious for everyone surely!).*

This is an interesting point. It is possible that those who persist with ABM find VS training more engaging. However we find that more participants complete at least one DOT ABM session than VS ABM – therefore the reverse may also be possible. Discussion on the asymmetric drop-off between conditions is included on (p.17, under *Limitations and future directions*), where the following has been added:

“Based on the number of participants reporting mood for at least six reports, it is also possible that participants who persist with ABM find VS training more engaging than DOT training.”

Minor point:

7. I did not find this statement (in the author contributions) clear: “Analysis was completed by AC and OJR who wrote the paper.” Does this mean that AC and OJR both wrote the paper or just OJR?

Thanks for highlighting this. The line has been updated to read the following for clarity:

“Analysis was completed by AC and OJR, both of whom wrote the paper.”